# Adherence to a Cholesterol-Lowering Diet and the Risk of Pancreatic Cancer: A Case–Control Study

**DOI:** 10.3390/nu16152508

**Published:** 2024-08-01

**Authors:** Matteo Di Maso, Livia S. A. Augustin, David J. A. Jenkins, Anna Crispo, Federica Toffolutti, Eva Negri, Carlo La Vecchia, Monica Ferraroni, Jerry Polesel

**Affiliations:** 1Department of Clinical Sciences and Community Health, Department of Excellence 2023–2027, Branch of Medical Statistics, Biometry and Epidemiology “G.A. Maccacaro”, Università degli Studi di Milano, via Celoria 22, 20133 Milan, Italy; matteo.dimaso@unimi.it (M.D.M.); monica.ferraroni@unimi.it (M.F.); 2Epidemiology and Biostatistics Unit, Istituto Nazionale Tumori—IRCCS—“Fondazione G. Pascale”, via M. Semmola 1, 80131 Naples, Italy; l.augustin@istitutotumori.na.it (L.S.A.A.); anna.crispo@istitutotumori.na.it (A.C.); 3Departments of Nutritional Science and Medicine, Temerty Faculty of Medicine, University of Toronto, Toronto, ON M5S 1A8, Canada; david.jenkins@utoronto.ca; 4Clinical Nutrition and Risk Factor Modification Centre, St. Michael’s Hospital, Toronto, ON M5C 2T2, Canada; 5Division of Endocrinology and Metabolism, Department of Medicine, St. Michael’s Hospital, Toronto, ON M5C 2T2, Canada; 6Li Ka Shing Knowledge Institute, St. Michael’s Hospital, Toronto, ON M5C 2T2, Canada; 7Unit of Cancer Epidemiology, Centro di Riferimento Oncologico di Aviano (CRO), IRCCS, via F. Gallini 2, 33081 Aviano, Italy; federica.toffolutti@cro.it (F.T.); polesel@cro.it (J.P.); 8Department of Medical and Surgical Sciences, Alma Mater Studiorum, Università di Bologna, via P. Palagi 9, 40138 Bologna, Italy; eva.negri@unibo.it; 9Fondazione IRCCS Ca’ Granda, Ospedale Maggiore Policlinico, via F. Sforza 35, 20122 Milan, Italy

**Keywords:** cholesterol, pancreatic cancer, plant-based diet

## Abstract

Background: Pancreatic cancer risk has been associated with increased serum cholesterol level, which is in turn partially influenced by diet. This study aimed at evaluating the association between pancreatic cancer risk and the adherence to a plant-based cholesterol-lowering diet. Methods: Data were derived from an Italian case–control study including 258 pancreatic cancer patients and 551 controls. The cholesterol-lowering diet score was based on seven components: high intakes of (i) non-cellulosic polysaccharides (a proxy of viscous fibers), (ii) monounsaturated fatty acids, (iii) legumes, and (iv) seeds/corn oils (a proxy of phytosterols); and low intakes of (v) saturated fatty acids, (vi) dietary cholesterol, and (vii) food with a high glycemic index. The score was calculated adding one point for each fulfilled component, thus ranging from zero (no adherence) to seven (complete adherence). The odds ratios (ORs) and 95% confidence intervals (CIs) were estimated through the logistic regression model. Results: Scores 5–7 were associated with reduced cancer risk (OR = 0.30; 95% CI: 0.18–0.52) compared to scores 0–2. Conclusions: Adherence to a plant-based cholesterol-lowering diet was associated with a reduced risk of pancreatic cancer.

## 1. Introduction

Pancreatic cancer is a fatal cancer affecting approximately 140,000 new patients in Europe, causing approximately 132,000 deaths each year [1]. In both sexes, pancreatic cancer ranked 14th for incidence (age-standardized incidence rate: 7.8/100,000), but 5th for mortality, with rates close to incidence (age-standardized mortality: 7.2/100,000) [1]. Since 2020, the incidence of pancreatic cancer has increased by about 27%, and a further increase is predicted in the near future [1].

Tobacco smoking is the major modifiable risk factor for pancreatic cancer [2]. However, the reduction in smoking prevalence had only a limited impact on pancreatic cancer trends, suggesting that other etiological factors are sustaining pancreatic cancer incidence. Among other modifiable factors, excessive body weight and dietary factors have been associated with increased risk of pancreatic cancer [3]. Studies on diet quality—measured through common a priori indices such as the Healthy Eating Index (HEI-2005 and HEI-2010), the Alternative Healthy Eating Index-2010 (AHEI-2010), or the Dietary Approaches to Stop Hypertension (DASH)—reported reduced risk of pancreatic cancer among people adhering to a healthy dietary pattern and increased risk in those eating according to a Western dietary pattern [4]. In particular, the favorable patterns are characterized by elevated consumption of fruits and vegetables and by low intake of animal fats. This was confirmed by a dietary modification trial of the Women’s Health Initiative [5], which found a 29% reduction in pancreatic cancer risk in overweight women following a low-fat diet.

However, total fat intake includes different types of fats, which can have different effects on cancer risk. Indeed, a high intake of saturated fatty acids (SFAs) [6,7,8,9] and trans-fatty acids [9,10] has generally been associated with an increased risk of pancreatic cancer, whereas monounsaturated fatty acids (MUFAs) and polyunsaturated fatty acids (PUFAs) have been associated with a reduced risk [7,8,11] or no association [6,9]. Evidence from pre-clinical studies supported the heterogeneity effect on pancreatic cancer risk according to type of fat [12]. Among dietary factors, special attention has been paid to cholesterol. Observational studies found a significant excess in the risk of pancreatic cancer associated with high intakes of dietary cholesterol [13], though this association was mainly found in case–control studies. Studies evaluating serum cholesterol reported similar results [14,15], but the comparison of results on dietary and serum cholesterol is challenging. Indeed, dietary cholesterol is only one contributor to serum cholesterol levels, which may explain the low correlations observed in some studies [16]. Nonetheless, circulating serum cholesterol levels are influenced by dietary habits, which are characterized by different sources of fats, cholesterol-lowering foods, and nutrients.

On this ground, a plant-based cholesterol-lowering Portfolio Diet Score has been recently developed for cardiovascular disease prevention [17]; this was adapted to the Italian diet to investigate the impact of cholesterol-lowering diets in cancer etiology [18] and validated on clinical trial data [19]. We hypothesized that adherence to a cholesterol-lowering diet is inversely associated with the risk of pancreatic cancer, consistent with previous findings for prostate cancer [18]. Specifically, the aims of this study were (a) to investigate the association between the cholesterol-lowering diet score and the risk of pancreatic cancer in an observational study conducted in Italy; and (b) to evaluate the potential modifying effects of common determinants of cancer risk (e.g., sex, age, tobacco smoking, alcohol drinking, overweight/obesity, physical activity).

## 2. Materials and Methods

### 2.1. Participants and Study Design

Data were derived from a multicentric case–control study conducted between 1991 and 2008 [7]. Cases were 326 patients (median age 63 years) with incident confirmed pancreatic cancer, admitted to major teaching and general hospitals in the province of Pordenone and in the greater Milan area, Northern Italy. For each case, two controls were frequency-matched by study center, sex, and age, for a total of 652 patients (median age 62 years). They were admitted to the same hospitals as cases for a wide spectrum of acute conditions other than neoplastic or digestive tract diseases, namely trauma (31%), other orthopedic disorders (31%), acute surgical conditions (28%), and other miscellaneous illnesses (10%). Refusal rate was below 5% for both cases and controls. In each of the participating centers, the local institutional authority reviewed the protocol and cleared the conduct of this study, in accordance with the Italian legislation for observational studies in force at the time this study was conceived. Patients signed an informed consent prior to enrollment. To avoid bias due to lipid-lowering medications, 68 cases and 101 controls treated for hypercholesterolemia were excluded from the present analysis, thus leaving 258 cases and 551 controls.

### 2.2. Diet Assessment and Cholesterol-Lowering Diet Score Computation

Trained interviewers administered a structured questionnaire to both the cases at the time of cancer diagnosis and the controls at hospital admission. The questionnaire assessed socio-demographic characteristics, anthropometric variables, smoking, drinking, physical activity, medical history, and history of cancer in relatives. Current smokers were patients who smoked at least 1 cigarette/day for a minimum continuous period of 1 year. Similarly, current alcohol users were patients who drank at least 1 drink/day for a continuous period of at least 1 year. Former smokers and alcohol users were individuals who had abstained from the habit for at least 1 year before diagnosis (for cases) or interviews (for controls). Overall physical activity, defined by the combination of occupational and recreational physical activities, was categorized as low, medium, and high [20].

The habitual diet during the two years prior to cancer diagnosis (for cases) or hospital admission (for controls) was assessed by a 78-item food-frequency questionnaire (FFQ), which had satisfactory validity (the median value of partial correlation coefficients between nutrient intakes assessed by the FFQ and a 7-day dietary record was 0.46) [21] and reproducibility (the median value of partial correlation coefficients between nutrient intakes assessed by two consecutive FFQs was 0.67) [22]. Participants were asked to indicate the average weekly frequency of consumption of each food item, reporting variation in seasonal consumption of fruit and vegetables. The serving size was defined in “natural” units (e.g., one egg, one apple) or as an average serving in the Italian diet (e.g., 80 g of dry pasta, 150 g of tomatoes). Intakes lower than once a week but at least once a month were coded as 0.5 per week. Total energy and nutrient intakes—including dietary fiber, MUFAs from animal and plant sources, and SFAs—were computed using an Italian food composition database [23]. Dietary fiber was derived using the Englyst method, assessing cellulose separately from soluble and insoluble non-cellulosic polysaccharides. Total MUFA and SFA intakes were then expressed as a proportion of total energy intake, multiplying the intake in grams by 9 kcal and dividing by total energy intake (in kcal). The glycemic index (GI) was expressed as a percentage of the glycemic response elicited by white bread as a standard food, using international tables [24]. The average daily GI of a subject’s diet was computed as the sum of the weighted GI values for each individual food weighted by weekly consumption [25].

The adherence to a cholesterol-lowering diet was assessed through an ad hoc score [18], validated on clinical data [19]. Briefly, the score considered seven a priori dietary components derived from the FFQ: (1) high intake of non-cellulosic polysaccharide soluble fibers (NCPSFs) as a proxy of high viscous fibers; (2) high MUFA intake; (3) high legume intake; (4) low SFA intake; (5) high intake of oils from seeds or corn as a proxy of phytosterols [26]; (6) low dietary cholesterol intake; and (7) low daily GI [27]. One point score was assigned for each of the seven dietary components, and zero elsewhere. The final score was calculated as the sum of all the point scores, and it ranged from 0 (no adherence) to 7 (complete adherence). Low and high intake defined by the current literature [28,29] were scarcely discriminating in the present study population; therefore, for each dietary component, a sex-specific cut-off was identified for women and men separately through a receiving operating characteristic (ROC) analysis predicting “case” status [18]; the point score was assigned accordingly.

### 2.3. Statistical Analysis

The original sample size of 326 cases and 652 controls (case-to-control ratio: 1:2) was calculated at the time this study was conceived to estimate an odds ratio of pancreatic cancer ≥ 1.5 for several dichotomous exposures (e.g., ever vs never tobacco smoking, diabetes), fixing the a priori probabilities α = 0.05 and β = 0.20 (power = 80%).

Odds ratios (ORs) and the corresponding 95% confidence intervals (CIs) for categories of cholesterol-lowering diet score (i.e., 0 to 2; 3 to 4; and 5 to 7) were computed through logistic regression models conditioned on sex. A model with minimum adjustments including terms for study center, age (<60, 60–64, 65–69, ≥70 years), education (<7, 7–11, ≥12 years), year of interview (1991–1999, 2000–2008), and daily total energy intake (<2029, 2029–2459; 2460–2984; ≥2985 kcal/day) was firstly estimated. Subsequently, to account for potential confounding, a fully adjusted model was estimated, including additional terms for smoking habits (never, former, current <20 cigarettes/day, current ≥20 cigarettes/day), drinking habits (never, former, current), diabetes mellitus, family history of pancreatic cancer in first-degree relatives, and physical activity (low/medium, high). Differences in the distribution of cholesterol-lowering diet score across strata of socio-demographic characteristics as well as lifestyle factors among controls were evaluated by means of the chi-square test of independence.

In addition, the ORs and 95% CIs for a 1-point increment in the cholesterol-lowering diet score were further estimated in the overall study sample and across strata of potential effect modifiers, such as socio-demographic characteristics (i.e., sex, age, education, period of enrollment) and lifestyle factors (i.e., tobacco smoking, alcohol drinking, physical activity) that might have an impact on diet or lipid profile. Diabetes mellitus was not considered given the small number of subjects reporting this condition. Heterogeneity across strata was assessed by means of the likelihood-ratio test comparing models with and without the interaction term. All tests were two-sided, and the significance level was set at 0.05. All analyses were conducted using R version 4.0.5.

## 3. Results

The distribution of socio-demographic characteristics and potential confounders is reported in Table 1. The cases reported high education, diabetes mellitus, and current smoking more frequently, whereas no differences emerged for other characteristics.

For each dietary component, the gender-specific optimal cut-off discriminating cases and controls is reported in Table 2. A reduced pancreatic cancer risk was found in people with high intakes of dietary fiber (NCPSFs) (i.e., ≥6.9 g/day for women and ≥7.3 g/day for men; OR = 0.43; 95% CI: 0.29–0.65) and in those with low glycemic index (i.e., <71.0 for women and <70.9 for men; OR = 0.40; 95% CI: 0.27–0.59). A non-significant association emerged for the remaining dietary components. Combining the seven dietary components, we evaluated the overall adherence to the cholesterol-lowering diet, as follows: high adherence, defined as a total score of 5 to 7, was reported in 17.1% of the cases and in 27.2% of the controls (Table 2), leading to an OR of 0.30 (95% CI: 0.18–0.52) compared to patients with 0–2 scores. Results were consistent when the patients with treated hypercholesterolemia were included in the analysis (Appendix A).

Among the controls, adherence to the cholesterol-lowering diet was significantly higher in women than in men (*p* = 0.02; Table 3) and in never-smokers compared to current and former smokers (*p* = 0.05), while no significant differences emerged for other socio-demographic characteristics or lifestyle factors.

The heterogeneity of the ORs was also investigated across strata of selected socio-demographic characteristics and lifestyle factors (Figure 1). Overall, an increment of one point in the score was associated with a significantly reduced risk of pancreatic cancer (OR = 0.72; 95% CI: 0.63–0.83). No significant heterogeneity emerged across strata of sex, age, education, and year of interview. The inverse association between pancreatic cancer risk and cholesterol-lowering diet score was confirmed in people with low/medium physical activity (OR = 0.62; 95% CI: 0.51–0.75), but not in those with high physical activity (OR = 0.94; 95% CI: 0.74–1.20; *p* for heterogeneity < 0.01). Similarly, the inverse association seems restricted to current alcohol users (OR = 0.66; 95% CI: 0.56–0.78), though the ORs were not significantly heterogeneous (*p* = 0.07).

## 4. Discussion

The present study reported an inverse association between the risk of pancreatic cancer and the cholesterol-lowering diet score. People with high adherence showed a reduction of 67% in pancreatic cancer risk compared to those with low adherence.

The reported association is strong and possibly overestimated. Nonetheless, it is likely to be genuine. The association between serum cholesterol and pancreatic cancer risk has been reported to vary according to the time to diagnosis [14,15]. Close to pancreatic cancer diagnosis (i.e., within three years), inverse associations have been found for high levels of serum cholesterol [15]. This is probably due to reverse causation: pancreatic cancer may reduce caloric intake and exert a “cholesterol-lowering effect” when it accelerates the cholesterol metabolism to build new membranes and maintain active signaling [30]. In contrast, high serum cholesterol levels were associated with an increased risk of pancreatic cancer when blood samples were taken more than four years prior to diagnosis [15]. This couples with the ability of cholesterol to promote pancreatic cancer cell proliferation, migration, and invasion [31]. Therefore, it is plausible that a diet that lowers circulating cholesterol levels may exert a protective effect towards pancreatic cancer onset. In the present study, we assessed the diet in the two years preceding enrollment, and this was representative of a longer period before diagnosis since the dietary habits were quite stable in the Italian population.

The results of the present study are consistent with previous investigations on similar a posteriori dietary patterns based on food consumption. A recent meta-analysis of six case–control studies and six cohorts reported a 24% increase in pancreatic cancer risk in people adhering to a Western-type dietary pattern [32], which is characterized by elevated intakes of red and processed meat, high-fat products, refined grains, high dietary GI, and low intakes of fruits and vegetables. These findings are supported by a diet modification intervention to reduce total fat to 20% of the daily energy intake in the Women’s Health Initiative, which showed a reduction of 29% in pancreatic cancer risk in overweight women in comparison to those who maintained their usual diet [5].

Hypercholesterolemia causes both innate and adaptive inflammatory responses [33], which may enhance the risk of pancreatic cancer [34]. This is in connection with the increased risk of pancreatic cancer in patients with type 2 diabetes mellitus. Similarly, acute pancreatitis, a medical inflammatory condition enhancing the risk of pancreatic cancer, has been linked to high serum cholesterol [35]. Interestingly, a cohort study found that some of the dietary components used in our cholesterol-lowering diet score (namely, low dietary cholesterol intake, low percentage of total calories from SFAs, and high intake of fiber and legumes) were associated with reduced risk of acute pancreatitis [36].

Beside dietary cholesterol, two other dietary components were significantly associated with reduced pancreatic cancer risk: low GI and high non-cellulosic polysaccharide soluble fibers. The association between GI and pancreatic cancer risk is however controversial since cohort studies thus far have generally failed to find any association [37]. Nevertheless, the beneficial effects of a low dietary GI were mostly seen in metabolically challenged compared to non-metabolically challenged subjects [38]. Indeed, a cohort study found a significant association between a dietary GI-related measure (i.e., the insulin load) and pancreatic cancer risk in people with pre-existing insulin resistance but not in insulin-sensitive subjects [39]. A recent prospective analysis of the Prostate, Lung, Colorectal, and Ovarian Cancer Screening Clinical Trial on the effect of selected screening examinations found an inverse association between a low dietary carbohydrate score and the risk of pancreatic cancer, which remained significant only in people ≥ 65 years of age (hazard ratio = 0.49, 95% CI: 0.34, 0.72) but not in younger people (hazard ratio = 1.03, 95% CI: 0.59, 1.81). These results suggest that insulin resistance is an effect modifier. Similarly, in our study, the association between the cholesterol-lowering diet score and pancreatic cancer risk was significant only in people leading a sedentary lifestyle (i.e., a risk factor for insulin resistance and obesity) but not in physically active subjects. Perhaps a significant effect of high GI has been seen in a recent publication on the association of GI in large cohorts with diabetes-related cancers, of which pancreatic cancer was one of the seven diabetes-related cancers [40]. Furthermore, a systemic review of randomized trials found that low GI lowers the level of low-density lipoprotein (LDL) cholesterol [27] through a reduced stimulation by insulin of HMG-CoA reductase, a rate-limiting enzyme of hepatic cholesterol synthesis. Intake of soluble fibers has also been inversely associated with the risk of pancreatic cancer in observational studies [41]. Viscous fibers—such as β-glucans, pectins, and gums—have been shown to hamper the movement of bile acids into the enteric micelles and the subsequent uptake of micelles into the enterocytes [42]. Consequently, there is a greater fecal bile acid loss that leads to a reduction in the enterohepatic bile acid circulation and an increased hepatic cholesterol utilization to replenish the liver bile acid pool.

The inverse association between adherence to a cholesterol-lowering diet and the risk of pancreatic cancer is consistent across strata of major risk factors, except for physical activity. In the present study population, physical activity was not associated with adherence to the cholesterol-lowering diet score; nonetheless, it could be an effect modifier. Indeed, physical activity may impact the lipid profile by reducing the total cholesterol level and increasing high-density lipoprotein (HDL) cholesterol [43]. Therefore, subjects reporting high physical activity may already have a favorable lipid profile and, thus, be less sensitive to the reduction in cancer risk due to the adherence to the cholesterol-lowering diet.

We acknowledge some potential study limitations. Information bias is possible due to several potential sources of bias. The cases may have recalled their dietary habits differently from controls, and the precision in reporting food consumption might depend on the frequency of consumption itself. Furthermore, recall bias is possible, considering that the usual diet was estimated over the two years prior to enrollment. To reduce these potential sources of bias, the questionnaire was administered to cases and controls by the same interviewers under similar conditions during their hospital stay. In addition, awareness of any hypotheses on dietary habits in the etiology of pancreatic cancer was limited in the Italian population at the time of this study. Furthermore, the FFQ was successfully tested for validity and reproducibility [21,22] in assessing dietary habits. Nevertheless, we cannot exclude the possibility of residual recall bias. This study enrolled cases and controls from 1991 to 2008, and some dietary habits may have changed in this 18-year period. However, dramatic dietary changes were unlike in the study period, since diet habits were quite stable in the Italian population of the same age as the study population. Nonetheless, if diet modifications occurred, they were within the Mediterranean diet style since the influence of ethnic diets was very limited at the time this study was conducted. Finally, the score was created by assigning one point when the dietary requirement was met and zero otherwise; this may have introduced misclassification since it did not weight each dietary indicator according to its impact on pancreatic cancer risk. However, differently from clinical predictive and prognostic scores, the intent of the cholesterol-lowering diet score was to evaluate associations with outcomes at a group level rather than at an individual level; therefore, potential classification bias is negligible. Notably, this approach is quite common and widely accepted in etiological and observational studies where the adherence to specific a priori dietary patterns was evaluated [44,45,46,47,48,49].

Although the interviewers were instructed not to include people with weight loss prior to cancer diagnosis, abdominal obesity was reported in 18.6% of the cases and 39.7% of the controls, suggesting possible misclassification; therefore, the analyses including obesity were prone to bias and the results could not be adjusted for or stratified by obesity. However, logistic regression models were adjusted for physical activity, which is often a good proxy for body weight. The similarity of results from the minimally and fully adjusted models further reassures us against any relevant role of residual confounding. Selection bias was also possible, though the cases and controls were enrolled from the same hospital catchment areas, and careful attention was paid to exclude from the control group subjects admitted for any condition associated with a long-term modification of dietary habits. Finally, the present score might merely mirror other scores of healthy dietary habits, such as the Mediterranean diet [50]. The Spearman correlation coefficient between the cholesterol-lowering diet score and the Mediterranean diet score was 0.35 (95% CI: 0.29–0.41), indicating a modest correlation. This indicates that the cholesterol-lowering diet score identifies a specific dietary pattern, which is different from other healthy diet patterns. With reference to confounding, we were able to allow for several potential confounders, including physical activity and diabetes.

A number of factors have to be considered in regard to the external validity of results. Firstly, this study was conducted in a geographic area where the Mediterranean diet was widespread, potentially limiting the external validity of the study findings to population eating according to different dietary styles. Secondly, food fortification with phytosterols was not considered since it was not in place at the time of diet assessment. However, food fortification is more widespread nowadays, and this aspect should be considered in nutrient estimations. Finally, although the cholesterol-lowering diet score is an a priori dietary pattern, its optimal cut-offs were determined on data and may be different in other contexts. However, the cut-offs in the present study were comparable to those of a similar study on prostate cancer [18]; the distribution of the cholesterol-lowering diet score among the control population of these studies was also similar, showing a stable dietary behavior.

The results of the present study are strengthened by a previous validation of the cholesterol-lowering diet score [19]. In a previous study on women with breast cancer (the DEDiCa trial), we showed an inverse association between the cholesterol-lowering diet score and serum levels of LDL cholesterol and total cholesterol. Notably, the proportion of patients within the recommended ranges of serum total cholesterol (i.e., <200 mg/dL), LDL cholesterol (i.e., <116 mg/dL), and triglycerides (i.e., <150 mg/dL) declined with increasing adherence to the cholesterol-lowering diet. Although the score was not validated on men yet, the DEDiCa study provided reassurance that, among women, the cholesterol-lowering diet score is associated with the lipid profile. Furthermore, the results are strengthened by the collection of extensive dietary information through a reproducible and valid FFQ [21,22], which was administered by trained interviewers through a direct interview.

## 5. Conclusions

In conclusion, adherence to a plant-based cholesterol-lowering diet is associated with decreased risk of pancreatic cancer, adding evidence on the beneficial effects of plant-based diets in reducing cancer risk. Physical activity is a potential effect modifier of this association, since the inverse association held true only among subjects with low/medium physical activity. The confirmation of this study’s findings in cohort studies would further support the favorable effect of plant-based diet in preventing pancreatic cancer.

## Figures and Tables

**Figure 1 nutrients-16-02508-f001:**
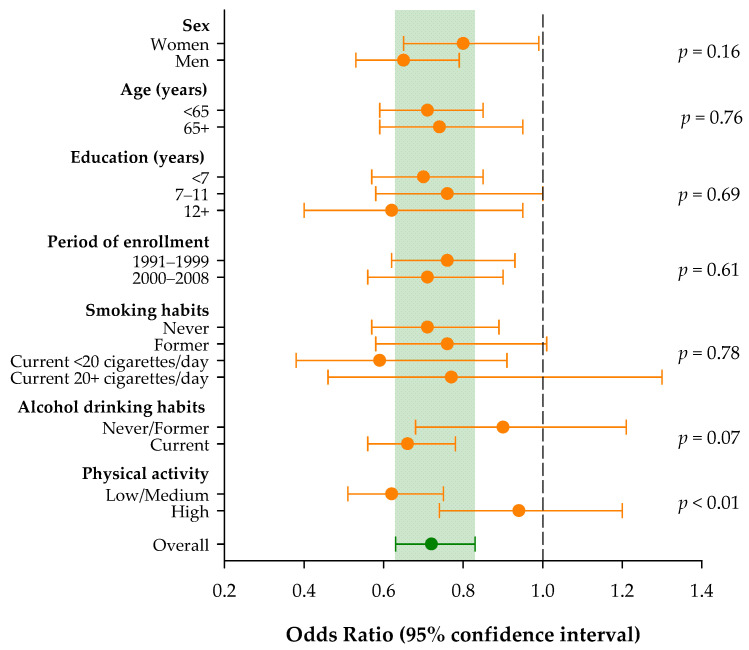
Odds ratios and corresponding 95% confidence intervals for pancreatic cancer risk according to a 1-point increment in the adherence to the cholesterol-lowering diet score and across strata of selected socio-demographic characteristics and lifestyle factors. Italy, 1991–2008.

**Table 1 nutrients-16-02508-t001:** Distribution of 258 cases of pancreatic cancer and 551 controls according to study center, socio-demographic characteristics, and lifestyle factors. Italy, 1991–2008.

Variable	Cases		Controls	
	*n*	(%)	*n*	(%)
Study center				
Pordenone	138	(53.5)	303	(55.0)
Milan	120	(46.5)	248	(45.0)
Sex				
Men	140	(54.3)	304	(55.2)
Women	118	(45.7)	247	(44.8)
Age (years)				
<60	102	(39.5)	212	(38.5)
60 to 69	93	(36.0)	191	(34.7)
≥70	63	(24.4)	148	(26.9)
Year of interview				
1991–1999	130	(39.9)	395	(71.7)
2000–2008	155	(60.1)	156	(28.3)
Education (years) ^a^				
<7	128	(49.6)	295	(53.5)
7 to 11	70	(27.1)	165	(29.9)
≥12	58	(22.5)	89	(16.2)
Physical activity ^a^				
Low/medium	143	(55.4)	342	(62.1)
High	115	(44.6)	207	(37.6)
Diabetes mellitus ^a^				
No	213	(82.6)	518	(94.0)
Yes	44	(17.1)	33	(6.0)
Family history of pancreatic cancer				
No	249	(96.5)	537	(97.5)
Yes	9	(3.5)	14	(2.5)
Smoking habits ^a^				
Never	110	(42.6)	271	(49.2)
Former	69	(26.7)	167	(30.3)
Current < 20 cigarettes/day	41	(15.9)	68	(12.3)
Current ≥ 20 cigarettes/day	37	(14.3)	43	(7.8)
Alcohol drinking habits ^a^				
Never	34	(13.2)	89	(16.2)
Former	26	(10.1)	48	(8.7)
Current	196	(76.0)	414	(75.1)

^a^ The sum does not add up to the total because of some missing values.

**Table 2 nutrients-16-02508-t002:** Adjusted odds ratios (ORs) and corresponding 95% confidence intervals (CIs) for pancreatic cancer risk according to dietary component ^a^ and cholesterol-lowering diet score. Italy, 1991–2008.

Dietary Component		Score	Cases		Controls		OR (95% CI) ^b^	OR (95% CI) ^c^
Women	Men		*n*	(%)	*n*	(%)		
Non-cellulosic polysaccharide soluble fibers (g/day), as a proxy of viscose fibers								
<6.9	<7.3	0	124	(48.1)	206	(37.4)	Ref.	Ref.
≥6.9	≥7.3	1	134	(51.9)	345	(62.6)	0.42 (0.28–0.62)	0.43 (0.29–0.65)
Monounsaturated fatty acids (% of total energy intake)								
<11.5	<12.0	0	47	(18.2)	153	(27.8)	Ref.	Ref.
≥11.5	≥12.0	1	211	(81.8)	398	(72.2)	1.43 (0.96–2.12)	1.46 (0.97–2.20)
Legumes (servings/week)								
<1	0	0	112	(43.4)	198	(35.9)	Ref.	Ref.
≥1	>0	1	146	(56.6)	353	(64.1)	0.74 (0.52–1.05)	0.80 (0.55–1.15)
Saturated fatty acids (% of total energy intake)								
≥11.2	≥9.7	0	143	(55.4)	280	(50.8)	Ref.	Ref.
<11.2	<9.7	1	115	(44.6)	271	(49.2)	0.76 (0.55–1.04)	0.80 (0.57–1.12)
Seeds or corn oil (g/day per 2000 kcal), as a proxy of phytosterol								
<3.1	<2.7	0	159	(61.6)	285	(51.7)	Ref.	Ref.
≥3.1	≥2.7	1	99	(38.4)	266	(48.3)	0.77 (0.54–1.08)	0.83 (0.58–1.18)
Dietary cholesterol (mg/day)								
≥303.7	≥235.1	0	161	(62.4)	290	(52.6)	Ref.	Ref.
<303.7	<235.1	1	97	(37.6)	261	(47.4)	0.59 (0.37–0.93)	0.62 (0.39–1.00)
Glycemic index ^d^								
≥71.0	≥70.9	0	206	(79.8)	353	(64.1)	Ref.	Ref.
<71.0	<70.9	1	52	(20.2)	198	(35.9)	0.43 (0.30–0.62)	0.40 (0.27–0.59)
Cholesterol-lowering diet score								
0 to 2			66	(25.6)	67	(12.2)	Ref.	Ref.
3 to 4			148	(57.4)	334	(60.6)	0.39 (0.26–0.60)	0.40 (0.26–0.63)
5 to 7			44	(17.1)	150	(27.2)	0.27 (0.16–0.45)	0.30 (0.18–0.52)

^a^ Cut-offs for each dietary component were selected by means of a receiving operating characteristic (ROC) analysis for men and women separately. ^b^ Estimated by means of logistic regression models conditioned on sex and adjusted for study center, age, year of interview, education, and energy intake. ^c^ Estimated by means of logistic regression models conditioned on sex and adjusted for study center, age, year of interview, education, physical activity, diabetes mellitus, family history of pancreatic cancer, smoking habits, drinking habits, and energy intake. ^d^ White bread scale.

**Table 3 nutrients-16-02508-t003:** Distribution of 551 controls according to cholesterol-lowering diet score and socio-demographic characteristics and lifestyle factors. Italy, 1991–2008.

Variable	Cholesterol-Lowering Diet Score						χ^2^ Test
	0 to 2		3 to 4		5 to 7		
	*n*	(%)	*n*	(%)	*n*	(%)	
Total	67	(12.2)	334	(60.6)	150	(27.2)	
Study center							
Pordenone	31	(10.2)	187	(61.7)	85	(28.1)	
Milan	36	(14.5)	147	(59.3)	65	(26.2)	*p* = 0.31
Sex							
Men	40	(13.2)	196	(64.5)	68	(22.4)	
Women	27	(10.9)	138	(55.9)	82	(33.2)	*p* = 0.02
Age (years)							
<65	35	(11.2)	191	(61.0)	87	(27.8)	
≥65	32	(13.4)	143	(60.1)	63	(26.5)	*p* = 0.71
Year of interview							
1991–1999	49	(12.4)	242	(61.3)	104	(26.3)	
2000–2008	18	(11.5)	92	(59.0)	46	(29.5)	*p* = 0.75
Education (years) ^a^							
<7	37	(12.5)	169	(57.3)	89	(30.2)	
7 to 1	21	(12.7)	102	(61.8)	42	(25.5)	
≥12	9	(10.1)	61	(68.5)	19	(21.3)	*p* = 0.39
Physical activity ^a^							
Low/medium	48	(14.0)	198	(57.9)	96	(28.1)	
High	19	(9.2)	134	(64.7)	54	(26.1)	*p* = 0.16
Diabetes mellitus ^a^							
No	64	(12.4)	308	(59.5)	146	(28.2)	
Yes	3	(9.1)	26	(78.8)	4	(12.1)	*p* = 0.08
Family history of pancreatic cancer							
No	64	(11.9)	327	(60.9)	146	(27.2)	
Yes	3	(21.4)	7	(50.0)	4	(28.6)	*p* = 0.52
Smoking habits ^a^							
Never	29	(10.7)	153	(56.5)	89	(32.8)	
Former	21	(12.6)	105	(62.9)	41	(24.6)	
Current < 20 cigarettes/day	10	(14.7)	42	(61.8)	16	(23.5)	
Current ≥ 20 cigarettes/day	6	(14.0)	33	(76.7)	4	(9.3)	*p* = 0.05
Alcohol drinking habits ^a^							
Never/Former	20	(14.6)	83	(60.6)	34	(24.8)	
Current	47	(11.4)	251	(60.6)	116	(28.0)	*p* = 0.53

^a^ The sum does not add up to the total because of some missing values.

## Data Availability

The data presented in this study are available for research purposes upon request from Jerry Polesel.

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
