# Peer review of "Adherence to a Cholesterol-Lowering Diet and the Risk of Pancreatic Cancer: A Case–Control Study"

_nutrients, 2024, doi:10.3390/nu16152508_

Round 1

Reviewer 1 Report

Comments and Suggestions for Authors

Dear Authors,

Congratulations on your valuable work, which, however, should be refined with regard to some aspects:

1) The objective is formulated too concisely. I suggest elaborating and clarifying it, perhaps with research questions and a research hypothesis.

2) In the ‘Introduction’ section, I propose also including trans isomers among etiological dietary factors (not only saturated fatty acids and cholesterol).

3) Please consider whether, for the sake of readability, section 2 (‘Materials and Methods’) could be sub-sectioned to detail participants, procedures, research tools and statistical analysis.

4) I suggest providing the validity and reliability coefficients of the implemented FFQ questionnaire,

5) In the ‘Statistical analysis’ section, please add the adopted p-value level.

6) Has the group size been statistically tested for a strong effect (e.g. G*Power)?

7) Please verify whether all of applied statistical tests are described in the ‘Statistical analyses’ section (e.g. in Table 3, the results of the chi-square test are shown).

8)Would it worth providing the p-value in Table 2?

9) Please verify the ‘Discussion’ so that all significant relationships that are part of the aim of the work are discussed against the background of the literature.

10) In my opinion, the ‘Conclusion’ is too concise. Moreover, it was not the purpose of the study to compare the risk of pancreatic and prostate cancers, thus, this comparison should be removed from the ‘Conclusion’ section.

Reviewer 2 Report

Comments and Suggestions for Authors

The paper is based on a case-control study conducted in a region of Milan in Italy and examines the adherence to a cholesterol-lowering diet and risk of pancreatic cancer. It provides evidence to suggest that there is an association of reduced pancreatic cancer risk in those adhering to cholesterol-lowering diet and suggested of effect modification with physical activity.

Some comments on the manuscript are provided to improve.

Abstract

In the results “physical activity was a potential effect modifier (p for heterogeneity <0.01), with OR=0.62 (95% CI: 0.51-0.75) and OR=0.94 (95% CI: 38 0.74-1.20) in physically active and inactive subjects, respectively”

This was a little confusing and requires some further explanation - is the interaction with the diet score and is this for association with 1 point increase or the categorised diet score 5+ vs 0-2 ?  It might be better to leave this out of the abstract.

Conclusion – ‘ adherence to a plant-based cholesterol-lowering diet decreased pancreatic cancer risk.’ The conclusion is perhaps too strong to say it ‘decreased’ as no causation is required. I think perhaps say there was an association between ..

Intro

“Over the last decades, incidence of pancreatic cancer has been increasing by about 27%” can you confirm in the paper exactly over what timeframe is this 27% referring to?

Methods

Cases and controls from one area in Milan – how representative are these of wider populations? Italy and meditarean diet – also the study from 1991-2008 – likely changes in diet over this 18 year period?

Exclusion of those with hypocholesterolaemia – could this have introduced bias (more exclusions as a proportion of cases than controls)?

When were individuals interviewed – diagnosis, treatment stage or follow-up /out patient?

Self-reported measures for lifestyle factors e.g. smoking, diet – likely introduced recall bias but this is not mentioned in limitations,

Drinker is not a term used – alcohol use or consumption might be used instead.

The FFQ is based on average weekly diet 2 years prior – this seems like a long window and recall might be an issue (no mention). Also overestimation of weekly amounts in some cases where at least monthly is counted as 0.5/week,

Ad-hoc score is quite a crude way of combining – it gives equal weight to each component. Might the authors have considered other weighting of items ?

No sample size or power calculations provided for associations considered.

Interactions are mentioned briefly at the end of the methods – which were considered and were they a priori determined? If so what evidence was available to support this e.g. physical activity appears to be an effect modifier but has this been shown previously?

Results

Table 2 legend should include ‘adjusted’ before OR.

What is the purpose of Table 3 – showing control group alone – this is not an aim or described in the methods and not certain of it’s place.

Figure 1: Selected sociodemographic variables included, but were not mentioned in the methods and why these specifically were selected ? evidence

Discussion

Line 235 ‘overweighted’ should be ‘overweight’

Line 238  use of the word ‘couples’ might be replaced.

Line 260 – there is no discussion of the interaction finding where low physical activity results in a bigger effect of diet on pancreatic risk – how is this explained, what does the finding mean?

Limitations – there is nothing on the time interval 1991-2008 and how diets may have changed over this time and what effect that might have had. Also the generalisability of the findings to other populations is likely to be limited to populations with similar diets.

The DeDica trial is mentioned as a validation study for the score and cholesterol levels but this trial was in women with breast cancer which does not represent the validation in the male population. This needs to be explicitly and that the validation here is not applicable to males.

Conclusion is too strong based on the limitations identified – this is a study of association only and that further studies are required to provide conclusive evidence.

Comments on the Quality of English Language

Some minor suggestions to improve quality of English. 

Reviewer 3 Report

Comments and Suggestions for Authors

This great article examines the link between nutrition and pancreatic cancer, a medical condition that is becoming more and more prevalent. The tables are informative, and the conclusions are appropriate. I do have some remarks:

Results: Why did you prefer the Bernouilli regression over the Poisson regression?

Discussion: The retrospective aspect of this study is a clear limitation, which should be discussed.

Comments on the Quality of English Language

Some typo's throughout the text.
